# Molecular Epidemiology and Baseline Resistance of Hepatitis C Virus to Direct Acting Antivirals in Croatia

**DOI:** 10.3390/pathogens11070808

**Published:** 2022-07-19

**Authors:** Petra Simicic, Anamarija Slovic, Leona Radmanic, Adriana Vince, Snjezana Zidovec Lepej

**Affiliations:** 1Department of Immunological and Molecular Diagnostics, University Hospital for Infectious Diseases, 10000 Zagreb, Croatia; psimicic@bfm.hr (P.S.); lradmanic@bfm.hr (L.R.); 2Centre for Research and Knowledge Transfer in Biotechnology, University of Zagreb, 10000 Zagreb, Croatia; aslovic@unizg.hr (A.S.); 3Department for Viral Hepatitis, University Hospital for Infectious Diseases, 10000 Zagreb, Croatia; avince@bfm.hr (A.V.); 4School of Medicine, University of Zagreb, 10000 Zagreb, Croatia

**Keywords:** hepatitis C virus, resistance associated substitutions, direct acting antivirals, epidemiology, phylogenetics, phylodynamics, genotypes, subtypes, intravenous drug users

## Abstract

Molecular epidemiology of hepatitis C virus (HCV) is exceptionally complex due to the highly diverse HCV genome. Genetic diversity, transmission dynamics, and epidemic history of the most common HCV genotypes were inferred by population sequencing of the HCV NS3, NS5A, and NS5B region followed by phylogenetic and phylodynamic analysis. The results of this research suggest high overall prevalence of baseline NS3 resistance associate substitutions (RAS) (33.0%), moderate prevalence of NS5A RAS (13.7%), and low prevalence of nucleoside inhibitor NS5B RAS (8.3%). Prevalence of RAS significantly differed according to HCV genotype, with the highest prevalence of baseline resistance to NS3 inhibitors and NS5A inhibitors observed in HCV subtype 1a (68.8%) and subtype 1b (21.3%), respectively. Phylogenetic tree reconstructions showed two distinct clades within the subtype 1a, clade I (62.4%) and clade II (37.6%). NS3 RAS were preferentially associated with clade I. Phylogenetic analysis demonstrated that 27 (9.0%) HCV sequences had a presumed epidemiological link with another sequence and classified into 13 transmission pairs or clusters which were predominantly comprised of subtype 3a viruses and commonly detected among intravenous drug users (IDU). Phylodynamic analyses highlighted an exponential increase in subtype 1a and 3a effective population size in the late 20th century, which is a period associated with an explosive increase in the number of IDU in Croatia.

## 1. Introduction

Molecular epidemiology of hepatitis C virus (HCV) infection is exceptionally complex due to the highly diverse viral genome, which is classified into >90 genotypes and subtypes [1]. HCV subtypes such as 1a, 1b, and 3a are considered to be epidemic and are widely distributed worldwide. They are characterized by lower variability and are considered to have spread across the world in the 20th century through unsafe medical procedures, transfusion, blood products, and intravenous drug use [2,3,4,5,6]. On the other hand, endemic subtypes are restricted to certain geographic areas such as the Middle East, South and East Asia, and West and Central Africa, have higher genetic variability, and are considered to be much older [5,6,7,8]. Furthermore, the existence of a quasispecies, which is described as distributions of genetically non-identical but related genomes subjected to a continuous process of competition and selection in a given host, is another factor contributing to the extensive HCV genetic diversity [9,10].

Croatia is a South-Eastern European country with a relatively low prevalence of HCV infection. It is estimated that 35,000–45,000 persons, or 0.9% of the Croatian population, are infected with chronic hepatitis C [11]. HCV genotype distribution in Croatia varies according to different population groups and regions. Genotype 1 (GT1) is the most prevalent among the general population (57–80%), followed by genotype 3 (GT3) (13–48%), while the prevalence of genotype 2 (GT2) (1–2%) and 4 (GT4) (4–7%) is low [11,12]. HCV transmission via blood transfusion and unsafe medical procedures was significantly reduced after the introduction of blood donation screening for HCV in the 1990s [11,13]. In Croatia, blood transfusion and haemodialysis before 1992 were strongly associated with an infection with HCV GT1, especially subtype 1b (GT1b) [11,14]. Today, intravenous drug use represents the major route of HCV transmission in developed countries, with a prevalence of 2–80% among intravenous drug users (IDU) [15,16,17]. Estimated population size of IDU in Croatia is around 15,000, and the most common HCV subtypes in this population are subtype 3a (GT3a) (61%) and subtype 1a (GT1a) (24%) [11,18].

The discovery of the new direct acting antivirals (DAA) has revolutionized the treatment of HCV patients and their efficacy has prompted the World Health Organization (WHO) to launch a strategy which calls for the elimination of viral hepatitis as a public health threat by 2030 [19]. Major changes have been implemented in HCV treatment strategies in Croatia in the last several years. Treatment eligibility was expanded to include patients aged 18–84 years, regardless of fibrosis stage, and the annual treated number of patients tripled from 2015 to 2018 [20]. However, some quasispecies variants bear polymorphisms in drug-targeted genes, which may negatively impact antiviral treatment and confer resistance to direct acting antivirals in 5–10% of the patients [21,22,23]. The use of direct acting antivirals with high barrier to resistance and a combination of several DAA classes can be beneficial in ensuring a successful achievement of sustained virological response (SVR). A combination of highly efficient pangenotypic DAA has reduced the need for baseline resistance testing [23,24,25,26]. However, current guidelines suggest testing for NS5A resistance associated substitutions (RAS) in the treatment of naïve patients with HCV GT3 and liver cirrhosis before treatment with sofosbuvir/velpatasvir since Y93H mutation was shown to reduce SVR to 84–88% in these patients [26,27]. Furthermore, testing for NS5A RAS is advised before treatment with grazoprevir/elbasvir in patients infected with HCV GT1a [26,28]. Literature data on baseline DAA resistance prevalence varies across different studies, partly due to inconsistent classification of observed substitutions as RAS. Our group previously conducted a preliminary analysis of baseline resistance to NS3 protease inhibitors and NS5A inhibitors in patients infected with GT1a and GT1, respectively. High prevalence of NS3 RAS (46.3%), especially Q80K RAS (42.6%), was found in GT1a-infected patients, while clinically relevant NS5A RAS were shown to be more common in GT1b (24.2%) compared with GT1a (7.8%) [29,30]. However, epidemic history and resistance prevalence to all DAA classes for the most common HCV genotypes and subtypes circulating in Croatia remain unknown. The aim of this study was to conduct a comprehensive analysis of the molecular, virological, clinical, and epidemiological characteristics of the HCV epidemic in Croatia in a four-year period.

## 2. Results

### 2.1. Study Population

Demographic, epidemiological, clinical, and laboratory data from 300 included patients are summarized in Table 1. The majority of patients were male (187/300, 62.3%) and had median HCV viral load 6.0 log_10_ IU/mL (5.5–6.3 log_10_ IU/mL). There were 109 GT1a-infected patients (36.3%) with median age 44.0 (40.0–50.5) years, 80 GT1b-infected patients (26.7%) with median age 61.0 (52.0–65.8) years, and 111 GT3a-infected patients (37.0%) with median age 43.0 (39.0–49.0) years. Fibroscan results were available for 293/300 included patients, with F0/F1 fibrosis stage being the most common (103/293, 35.2%). A similar number of patients had F2 (72/293, 24.6%) and F4 fibrosis (74/293, 25.3%), while 44 had F3 fibrosis (15%). Overall, reported HCV infection routes were intravenous drug use in 95 (31.7%) patients, iatrogenic in 48 (16.0%) patients, perinatal or sexual in 11 (3.7%) patients, while presumptive mode of HCV transmission was unknown for 146 (48.7%) patients. The most commonly reported HCV infection route was iatrogenic for patents infected with HCV subtype 1b (25/80, 31.3%) and intravenous drug use for patients infected with HCV subtype 1a (48/109, 44.0%) or 3a (46/111, 41.4%) (Table 1).

### 2.2. Resistance Analysis

The overall prevalence of baseline RAS in at least one DAA target was 45.7% (137/300). RAS were detected in 80 GT1a-infected patients (80/109, 73.4%), 48 GT1b-infected patients (48/80, 60.0%), and 9 GT3a-infected patients (9/111, 8.1%). Prevalence of resistance-conferring RAS in at least one DAA target was 27.7% (83/300). Single or multiple RAS from resistant class were detected in 60 GT1a-infected patients (60/109, 55.0%), 19 GT1b-infected patients (19/80, 23.8%), and 4 GT3a-infected patients (4/111, 3.6%). Differences in patient characteristics by NS3, NS5A, and NS5B RAS are summarized in Table 2. Comparison of selected demographic, epidemiological, clinical, and laboratory data showed no statistically significant association between prevalence of NS3, NS5A, or NS5B RAS and patients’ gender, age, fibrosis stage, or HCV viral load.

#### 2.2.1. Prevalence of NS3 Specific RAS

At least one baseline NS3 RAS was detected in 32.7% (98/300) patients with the prevalence of clinically relevant substitutions of 20.7% (62/300). Prevalence of NS3 RAS significantly differed according to HCV genotype, with NS3 RAS being the most common in GT1a (75/109, 68.8%), followed by GT1b (21/80, 26.3%), and lastly, GT3a (2/111, 1.8%) (*p* < 0.001). Similarly, resistance-conferring RAS were dominant in GT1a-infected patients (57/109, 52.3%) but rarely present in patients infected with HCV GT1b (4/80, 5.0%) and GT3a (1/111, 0.9%) (Figure 1a). The most commonly detected NS3 RAS in GT1a was Q80K with a baseline prevalence of 46.8% (51/109). Other clinically relevant NS3 RAS in GT1a were T54S (4/109, 3.7%) and V55A (8/109, 7.3%). Among RAS which confer reduced susceptibility to at least one NS3 inhibitor, N174S was relatively common (43/109, 39.5%). In patients infected with GT1b, only a few resistance-conferring RAS were detected: T54S (2/80, 2.5%), V55A (1/80, 1.3%), and N174F (1/80, 1.3%). Much more common was Y56F RAS (18/80, 22.5%) which causes reduced susceptibility to grazoprevir. In GT3a, only Q168R (1/111, 0.9%) and Q168K (1/111, 0.9%) were detected (Table 3).

#### 2.2.2. Prevalence of NS5A Specific RAS

At least one baseline NS5A RAS was detected in 13.7% (41/300) patients with the prevalence of clinically relevant substitutions of 8.0% (24/300). Prevalence of NS5A RAS also differed significantly according to genotype, with RAS being more common in GT1a (17/109, 15.6%) and GT1b (17/80, 21.3%) compared to GT3a (7/111, 6.3%) (*p* = 0.007). However, resistance conferring NS5A RAS were more prevalent in GT1b (16/80, 20.0%) compared to Gt1a (5/109, 4.6%). Overall, NS5A RAS were rarely detected in GT3a-infected patients, especially clinically relevant substitutions (3/111, 2.7%) (Figure 1b). The most commonly detected NS5A RAS in GT1a was M28V with a baseline prevalence of 10.1% (11/109). Clinically relevant NS5A RAS were Q30R (3/109, 2.8%), M28T (1/109, 0.9%), and L31M (1/109, 0.9%). Y93H was the primary mutation detected in GT1b-infected patients (9/80, 11.3%), with other resistance-conferring RAS being R30Q (5/80, 6.3%) and L31M (4/80, 5.0%). In GT3a-infected patients, RAS A62L (4/111, 3.6%), Y93H (2/111, 1.8%), and A30K (1/111, 0.9%) were detected (Table 4).

#### 2.2.3. Prevalence of RAS Specific for NS5B Nucleoside Inhibitors (NI)

Within NS5B, the prevalence of NI NS5B RAS was observed in 8.3% (25/300) of patients. All RAS were detected exclusively in GT1b-infected patients (25/80, 31.3%, *p* < 0.001), with the single observed variant being L159F which causes a reduced susceptibility to sofosbuvir. No NI NS5B RAS were observed in GT1a- and GT3a-infected patients.

#### 2.2.4. Prevalence of Multiple RAS

The overall prevalence of multiple RAS in different nonstructural regions was 8.3% (25/300). In 23 patients (23/300, 7.6%), dual class RAS were observed, while 2 patients (2/300, 0.7%) exhibited triple class RAS. A combination of multiple RAS in the NS3 and NS5A region was the most common (15/300, 5.0%), while NS3 + NI NS5B RAS (5/300, 1.7%) and NS5A + NI NS5B RAS (3/300, 1.0%) were rarely detected.

### 2.3. Phylogenetic Analysis

Maximum likelihood phylogenetic tree of the first dataset confirmed that all 300 Croatian sequences were correctly subtyped (Figure 2). The results of NS3 phylogenetic analyses for HCV GT1a, GT1b, and GT3a are shown as cladograms in Figure 3 and Appendix A. Two distinct clades of GT1a were observed: clade I with 62.4% (68/109) of Croatian GT1a sequences and clade II with 37.6% (41/109) of Croatian GT1a sequences. NS3 RAS were more prevalent in clade I (54/68, 79.4%) compared to clade II (21/41, 51.2%) (*p* = 0.003) (Table 2). Similarly, higher prevalence of resistance conferring NS3 RAS was found among clade I-infected patents compared to clade II-infected patients (51/68, 75.0% vs. 6/41, 14.6%, *p* < 0.001). The majority of clade I-infected patients (50/68, 73.5%) and only a single clade II-infected patient (1/41, 2.4%) had Q80K RAS. Subclades could not be observed for GT1b and GT3a. Croatian sequences were dispersed within background sequences from various countries regardless of RAS presence. The results of NS5A (Appendix A) and NS5B (Appendix A) phylogenetic analyses for all 300 patients’ sequences were concordant with phylogenetic analysis of the NS3 genomic region for GT1a, GT1b, and GT3a, which suggests the absence of recombination events between these regions. No statistically significant associations between GT1a clades and NS5A or NS5B RAS were observed (Table 2). Four transmission clusters with 8 patients (8/109, 7.3%) were consistently identified across all genomic regions for GT1a, one transmission cluster with 2 patients (2/80, 2.5%) for GT1b, and eight transmission clusters with 17 patients (17/111, 15.3%) for GT3a.

#### Transmission Cluster Characteristics

Transmission clusters were observed independently of RAS in the NS3, NS5A, or NS5B region (Table 2). However, they were notably more common among younger patients (median age = 37.0 years) compared to unclustered patients who had higher median age (46.0 years) (*p* < 0.001). Prevalence of transmission clusters significantly differed according to HCV genotype, with sequences obtained from GT3a-infected patients more commonly forming clusters and pairs (17/111, 15.3%) (*p* = 0.008). GT1a-infected patients with presumed epidemiologically linked infection were all part of clade I (8/68, 11.8%) (*p* = 0.024). Transmission clusters were more common among patients who reported IDU as a presumptive mode of transmission of HCV infection (14/95, 14.7%) compared to other HCV infection routes (13/192, 6.3%) (*p* = 0.018). No statistically significant association between clustering and patients’ gender, fibrosis stage, and HCV viral load was observed (Table 5).

### 2.4. Phylodynamic Analysis

Concatenated NS3, NS5A, and NS5B sequences (total length 2637 bp) were used to estimate GT1a, GT1b, and GT3a population dynamics. The substitution rate and time to most recent common ancestor (tMRCA) was inferred for every subtype group using different demographic and molecular clock models along with approximate marginal likelihoods (Table 6).

Bayes factor (BF) analysis favoured the relaxed lognormal molecular clock and the Bayesian skyline demographic model for all subtypes which were thus used for subsequent analysis (Appendix A). The mean estimated substitution rates of the GT1a, GT1b, and GT3a sequences were 1.48 × 10^−3^ (95% HPD, 0.52–2.40 × 10^−3^), 3.78 × 10^−3^ (95% HPD, 1.30–6.59 × 10^−3^), and 2.04 × 10^−3^ (95% HPD, 0.47–3.67 × 10^−3^), respectively. The mean estimates for the tMRCA of the analysed sequences were 1960 (95% HPD, 1911–1994) for subtype 1a, 1994 (95% HPD, 1973–2009) for subtype 1b, and 1978 (95% HPS, 1933–2004) for subtype 3a (Table 6). The Bayesian skyline plots (BSP) of the analysed dataset are shown in Figure 4. The skyline profile of the GT1a dataset shows a steady nonexpanding phase until the 1990s followed by an exponential increase in HCV infections from the 1990s to mid-2000s. Similarly, GT3a BSP shows expansion of the population size since tMRCA in the late 1970s and its stabilisation in the last decade. GT1b BSP shows a relatively constant population size from tMRCA in the early 1990s to the present time with a slight increase in the first decade of the 21st century.

## 3. Discussion

In this study, we combined molecular, virological, clinical, and epidemiological data from 300 HCV patients receiving clinical care at the University Hospital for Infectious Diseases Zagreb (UHID) and the Reference Centre for Viral Hepatitis in Croatia from 2016 to 2019, in order to characterize Croatian HCV epidemic. Previous studies of HCV epidemiology in Croatia were mainly focused only on the prevalence and distribution of various HCV genotypes and subtypes, while this study broadens this knowledge by including transmission and phylodynamic analyses. The most common HCV genotypes in the Croatian general population were found to be GT1 and GT3, with no major changes in molecular epidemiology in the last 20 years [12,31]. In this study, we observed higher median age of GT1b-infected patients (61 years) compared with GT1a (43 years) and GT3a (44 years), which is similar to data from a previous national study and could be attributed to predominantly iatrogenic mode of HCV transmission in GT1b which was significantly reduced in the last 30 years [11,31]. However, median ages of all patients included in this study were overall at least 10 years higher which suggests aging of the patient population and could be attributed to long persistence of HCV infection and the relatively recent availability of the effective direct antiviral therapy. The results of previous European and national studies are concordant with our finding that GT1a and GT3a were more frequently associated with IDU [3,11,17,32].

We conducted the first comprehensive analysis of baseline RAS across HCV genome of the most common genotypes circulating in Croatia. The prevalence of RAS varied greatly according to HCV genotypes. The highest prevalence of RAS was observed in the NS3 region (33.0%), especially among GT1a-infected patients (68.8%), with the most commonly detected RAS being Q80K (46.8%) and N174S (39.5%). Preliminary study conducted by Grgic et al. (2017) in the NS3 region of 136 GT1a-infected patients showed similar prevalence of Q80K and N174S RAS [29]. The prevalence of NS3 RAS was shown to be much lower in GT1b (5.0%) and GT3a (0.9%), especially when the analysis was limited to clinically relevant RAS. Literature data from other studies in Southern and Eastern Europe is limited. One Italian study showed RAS prevalence of 20.4% in the NS3 region, with the majority of NS3 RAS observed in GT1a (45.2%) and GT1b (10.8%). Similarly, to our study, most frequently observed RAS was Q80K in GT1a (17.0%) [33]. Q80K RAS was common in other European studies. Jimenez-Sousa et al. found prevalence of Q80K in the range from 7.3% to 22.2% in 2568 patients with chronic hepatitis C in 115 hospitals in Spain [34]. Beloukas et al. (2015) studied the occurrence of Q80K in 238 treatment-naïve HCV-1a carriers in England and found 14.9–27.1% prevalence of Q80K RAS [35]. It should be noted that Q80K mutation is considered clinically relevant in respect to simeprevir, which is first generation protease inhibitor. Today, most of the DAA combinations include third generation protease inhibitors such as grazoprevir, glecaprevir, and voxilaprevir, which show higher genetic resistance barrier and better pangenotypic activity [26,36,37]. However, Q80K RAS is still relevant in the context of new protease inhibitors since it causes reduced susceptibility to voxilaprevir [22,38]. In our study, Y56F RAS, which causes reduced susceptibility to grazoprevir, was common in GT1b (22.5%), followed by T54S (2.5%), V55A (1.3%), and N174F (1.3%). In Italy, several studies found very low prevalence of Y56F in GT1b (0–0.2%), while the most common RAS was T54S (1.9–4%) [33,39].

Results of this research showed higher prevalence of baseline NS5A RAS in GT1 (18.0%) compared with GT3a (6.3%). Furthermore, even though the overall prevalence of RAS was similar among different subtypes of GT1, a large difference was observed in the prevalence of clinically relevant RAS with 20.0% of GT1b-infected patients and 4.6% of GT1a-infected patients having resistance conferring mutations. Such discrepancy was first observed in our preliminary study where Y93H was the only RAS detected in GT1b-infected patients (24.2%) [30]. Current analysis revealed other RAS in GT1b-infected patients with the prevalence >5% (R30Q, L31M), while the prevalence of Y93H was lower (11.3%), even though it still represents the most common NS5A RAS. In this study, Y93H RAS was not detected in GT1a, while its prevalence in GT3a-infected patients was very low (1.8%). NS5A represents the most important genomic region for resistance testing since it contains the largest number of possible mutation sites, while many of the most commonly prescribed DAA combinations include NS5A inhibitor [38,40]. There is no recent data on the prevalence of resistance to NS5A inhibitors in south eastern Europe. An Italian observational study which included 1032 treatment naive patients from 23 clinical centres found NS5A RAS prevalence of 6.8% in GT1a, 10.3% in GT1b, and 8.5% in GT3a when the analysis was limited to clinically relevant RAS [33]. Other Italian studies found similar prevalence of NS5A RAS in GT3a (11.5%), GT1a (4.9%), and GT1b (23.0%) [41,42], while national studies across Europe found NS5A RAS prevalence of 2–18.9% in GT1a, 13.0–43.3% in GT1b, and 3–23.5% in GT3a [40,43,44,45,46]. It should be noted that such high variation in RAS prevalence among a single genotype could at least partly be attributed to the lack of standardisation in definition of RAS.

For the first time, this study analysed the prevalence of resistance associated substitutions to nucleoside NS5B inhibitors in Croatia. High genetic barrier to resistance and pangenetic activity of NS5B inhibitors are possible due to the highly conserved NS5B active site [23]. The only NI NS5B RAS found in the current study was L159F in GT1b (31.3%). Literature data on the prevalence of NS5B resistance in national studies are scarce. Several studies in southern Europe found similar prevalence of NI NS5B RAS in GT1b (14.8–21.1%) [33,40]. No NI NS5B RAS were observed in GT1a or GT3a patients [33,41].

Since every DAA class induces specific mutation profile characteristic for various HCV genotypes and subtypes, the use of combination therapy with agents targeting various viral proteins has been successful in obtaining high rates of SVR [23,24,25,26]. Only a few studies analysed the presence of RAS to all three DAA classes for every patient, mainly due to the lack of sequences spanning across all relevant genome regions. Our data show that multiclass resistance is relatively uncommon in Croatia (8.3%), with the most frequently observed RAS in both the NS3 and NS5A region (5.0%). The prevalence of combination RAS involving the NS5B region was low (<2%), especially NS3 + NS5A + NS5B RAS (0.7%). Global analysis of DAA RAS using published GenBank data similarly found low prevalence of multiple RAS in different regions of the same sequence (1.2–3.5%), with the exception of NS3 + NS5A RAS (15.6%) [47]. In a study conducted as a part of the AVIATOR, a phase 2 clinical trial, none of the patients had baseline RAS in all three targets [48]. Among European national studies, authors of a Portuguese study found 4.9% prevalence of NS5A + NS5B RAS, while an Italian study found 7.3% prevalence of multiclass RAS, with RAS simultaneously observed in the NS3 + NS5A region (2.7%), NS3 + NS5B region (1.9%) and NS5A + NS5B region (1.6%) [40]. The effect of the presence of multiple resistance substitutions is not extensively studied, however, it should be considered when administering DAA combinations which involve NS3 and NS5A inhibitors [33].

High molecular diversity of HCV genome due to continuous process of genetic variation enables the detection of evolutionary and epidemiological patterns in a relatively short time span [9,32,49,50,51]. Phylogenetic analyses of NS5A, NS5B, and NS3 sequences showed consistent separation of GT1a sequences into two distinct clades: clade I (62.4%) and clade II (37.6%). Proportion of clade I sequences seems to be slightly higher compared to previous preliminary national studies based on single gene phylogenies [29,30]. The prevalence of NS3 RAS was shown to be associated with clade, with NS3 RAS observed more often in clade I (79.4%) compared with clade II (51.2%) (*p* = 0.003). This phenomenon was even more pronounced when taking into consideration only clinically relevant RAS (75.0% clade I vs. 14.6% clade II, *p* < 0.001), especially Q80K mutation (73.5% clade I vs. 2.4% clade II, *p* < 0.001). This divergence of GT1a sequences in two distinct clades and preferential association of Q80K RAS in NS3 with clade I is in accordance with previous studies [34,48,52,53,54]. Picket et al. showed the existence of many clade-informative sites within the NS3, NS5A, and NS5B non-structural coding regions using full-genome sequencing data [52]. Krishnan et al. found that the majority GT1a sequences with Q80K RAS classified into clade I (98%), with overall prevalence of Q80K in clade I of 66.4% [48]. Overall prevalence of Q80K was 46.8% in the United States and 13.5% in the European Union [48]. Jimenes Souisa et al. also confirmed a higher prevalence of Q80K RAS in patients infected with clade I GT1a (41.5%) compared to clade II (1.6%) (*p* <0.001) [34]. De Luca et al. (2015) observed a significant difference in clade prevalence according to geographic origin, with higher prevalence of clade I among non-European sequences, represented mostly by sequences from the United States, compared with European sequences (75.7% vs. 49.3%; *p* < 0.001) Q80K RAS was detected exclusively in clade I sequences with prevalence of 51.6% [54]. Several studies suggest the origin of the Q80K polymorphism in the United States in clade I in the middle of the 20th century [53,54]. Our study shows high prevalence of clade I and Q80K RAS, which is comparable with data from the United States, suggesting efficient transfer of this polymorphism from the United States to Europe.

Sequence analysis of fast evolving viruses such as HCV was shown to be efficient method for viral transmission tracing [16,17,49,51]. Phylogenetic analysis demonstrated that 27 (9.0%) of Croatian HCV sequences had a presumed epidemiological link with another sequence and classified into 13 transmission pairs or clusters which were the most common in GT3a, while IDU was shown to be a risk factor statistically associated with clustering. Several studies performed phylogenetic cluster analysis among IDU to infer epidemiologic links and factors associated with clustering. Hackman et al. (2020) found that 46% of 820 community recruited IDU in Baltimore had genetically linked HCV infections with an average cluster size of 2–3 patients [49]. Clipman et al. (2021) analysed 483 HCV sequences from IDU in India and found transmission cluster prevalence of 28.8% with a median cluster size of three individuals [16]. Cluster analysis showed that younger age (<35 years) was significantly associated with being in a cluster, similarly to our study [16,50]. A study by Parczewski et al. (2018) showed that sequences obtained from patients with F3-F4 liver fibrosis less commonly formed clusters and pairs (22.2% vs. 43.7%, *p* < 0.001) and that NS5A RAS were less frequent among clustered sequences (5.2% vs. 11.2%, *p* = 0.039) [55]. However, in our research, fibrosis stage and presence of RAS showed no association with clustering. Palladino et al. (2020) showed similar prevalence of transmission clusters (10.5%) in the HCV GT1a population in Spain based on the NS5A region. However, male-dominated transmission pairs and predomination of clade II viruses were shown to be specific for the Spanish HCV-GT1a population [51]. In our research, the transmission pairs and clusters comprised exclusively of clade I viruses in GT1a, while the patients’ gender showed no association with clustering.

Phylodynamic analysis was performed on concatenated NS3, NS5A, and NS5B sequences of every patient in order to increase the confidence of inferred evolutionary relationships. Several studies have tried to reconstruct the origin and evolutionary history of the most common epidemic HCV subtypes such as GT1a, GT1b, and GT3a [4,32,50,56,57,58,59]. A study by Margiokinis et al. (2009) supports a massive expansion of the GT1a and GT1b epidemics between 1940 and 1980, with the expansion of HCV GT1b preceding that of GT1a by 15–17 years. Authors concluded that the global epidemic of both subtypes coincides with the vast increase of unsafe medical procedures during and after World War II, while GT1a expansion was at least in part driven by the increase of IDU after 1960 [4]. Cuypers et al. (2017) found that the first introduction of GT1a in Italy was timed around 1958 (95% HPD, 1949–1964), probably on multiple occasions from the US and Western Europe [56]. Our study shows that last common ancestor of Gt1a sequences dates around 1960 (95% HPD, 1911–1994), a period of the post-World War II and Cold War–era migrations. Hoshino et al. (2018) found the increase of GT1a population in Okinawa, Japan in two periods—from 1965 to 1980, which could be attributed to the US occupation after World War II and in the beginning of the 21st century which could be associated with an increase in the illicit drug use [59]. The Bayesian skyline analysis of Croatian GT1a sequences revealed an exponential increase of HCV GT1a infections in the 1990s and its continued growth throughout the first decade of the 2000s, which is in accordance with a huge increase in the number of IDU registered in this period in Croatia [60]. Unlike many other studies [4,32,50], the most recent common ancestor of Croatian GT1b sequences dates to the 1990s (95% HPD, 1975–2011), which is a period of introduction of HCV antibody screening of all donor blood and subsequent reduction in iatrogenic transmission of HCV characteristic for GT1b. The Bayesian skyline analysis suggests that GT1b population in Croatia was relatively constant in the last 30 years. However, the results have to be taken with caution due to the broad estimation errors (95% HPD limits). Furthermore, the tMRCA for GT1b sequences seems to be too recent to fit epidemiological data. Further research is needed in order to infer epidemiologic history of GT1b in Croatia.

Phylodynamic and phylogeographic analyses of GT3a epidemic history suggest that the most probable origin of this subtype is the Golden Crescent and Indian subcontinent [32,57]. Recent Montenegrin studies showed that GT3a was most likely exported from this area, which has a long history of opium production, to Europe in the first half of the 20th century by illicit drug trade, and reached Montenegro in the 1960s or 1970s, followed by exponential increase of effective population size [32,58]. The results of our research show that the last common ancestor of GT3a sequences in Croatia dates to 1978 (95% HPD, 1933–2004), causing an epidemic which exponentially grew until the last decade, suggesting similar temporal pattern. However, continuously high upper limit of 95% HPD prevents any definitive conclusions. The observed epidemic growth coincides with the huge increase in the number of new registered addicts in the early 1990s in Croatia [60]. The rise in illegal drug use in Croatia, including heroin, is considered to have begun in the 1960s, but became a pronounced social problem during the 1980s [60]. The first harm reduction programs were implemented in the early 2000s, and their efficiency could be partly observed in HCV GT1a and GT3a dynamics, with the effective population size of both subtypes being in a nonexpanding phase in the last decade.

There are some important limitations to this study. All samples were obtained from patients in the chronic phase of HCV infection; therefore, it is possible that the obtained sequences do not necessarily reflect true composition of viral quasispecies at the time of HCV transmission due to high variability of HCV genome. However, due to the lack of symptoms, HCV infection is rarely diagnosed in the acute phase and such samples are not readily available. Our research was conducted on sequences from three key non-structural regions of the HCV genome in order to cover DAA resistance associated sites, while the most accurate phylogenetic relationships would have been obtained by whole genome sequencing. We used Sanger sequencing which is a standard methodology used in molecular epidemiology studies. It allows detection of variants present in >15% of virus quasispecies which is sufficient for resistance analysis, while deep sequencing could possibly be beneficial for more confident tracing of viral transmission events and HCV genotype diversity, including mixed infections. Due to the lack of the patient follow-up, the effect of specific RAS on achievement of sustained viral response could not be evaluated. Furthermore, the concept of HCV compartmentalization suggests that viruses sampled from the blood at a given time point do not fully represent HCV infection dynamics [15]. However, the use of additional sites such as liver biopsies is ethically not acceptable since the successful implementation of non-invasive methods for assessing liver fibrosis. Our study covered the time span of four years which is a relatively short period. The accuracy of coalescent analysis could be improved by increasing the time frame of sampling. The presumptive mode of HCV transmission was unknown for 48.7% patients which limited the analysis of risk factors associated with HCV transmission. However, we remain confident that the results of this study provide useful and novel insights for tracing HCV genetic diversity, transmission dynamics, and epidemic history in Croatia.

## 4. Materials and Methods

### 4.1. Study Population

This cross-sectional retrospective study included 300 consecutive patients with chronic hepatitis C receiving clinical care at the UHID which is a Reference Centre for Viral Hepatitis in Croatia, in the period from January 2016 to December 2019. The inclusion criteria were as follows: age ≥ 18 years, viral load >1000 IU, HCV GT1a, GT1b, or GT3a. All patients qualified for HCV treatment, and none were previously treated with DAA. Selected demographic, epidemiological, clinical, and laboratory data (gender, age, reported HCV infection route, fibrosis stage, viral load, HCV subtype) were collected for every patient, when available. Fibrosis stage was previously assessed by noninvasive transient elastography (FibroScan; Echosens, Paris, France), while viral load and HCV genotype were determined by the Abbott RealTime HCV assay (Abbott, Abbott Park, IL, USA) and VERSANT HCV Genotype 2.0 Assay LiPA (Siemens Healthineers AG, Erlangen, Germany), respectively.

### 4.2. Amplification and Sequencing of HCV NS3, NS5A and NS5B Region

Viral RNA was extracted by QIAamp Viral RNA Mini Kit (Qiagen, Hilden, Germany) from 140 μL of serum, according to the manufacturer’s instructions. Amplification of the whole NS3 protease region (codons 1–181, 543 bp, nucleotides 3420–3962 of the reference strain H77, *GenBank* accession number: AF009606), domain I of NS5A region (codons 1–213, 639 bp, nucleotides 6258–6896 of the reference strain H77, *GenBank* accession number: AF009606), and partial NS5B region (codons 38–522, 1455 bp, nucleotides 7713–9167 of the reference strain H77, *GenBank* accession number: AF009606) was performed for each patient in two successive PCR reactions. In the first PCR reaction, viral RNA was reverse-transcribed, and cDNA was amplified using the Superscript^TM^ III One-Step RT-PCR System with Platinum^®^ Taq High Fidelity Polymerase Kit (Invitrogen, Waltham, MA, USA) with outer primers specific for each HCV subtype and genome region (Appendix A). Briefly, reaction was performed in 25 μL reaction tube with 12.5 μL 2× Reaction Mix, 0.5 μL forward primer (50 μM), 0.5 μL reverse primer (50 μM), 0.5 μL enzyme mix SuperScript™ III Reverse Transcriptase and Platinum^®^ Taq High Fidelity DNA Polymerase, 6.5 μL water, and 5 μL extracted HCV RNA. Reaction mixtures were incubated for 30 min at 50 °C followed by initial denaturation for 2 min at 94 °C and 45 cycles at 94 °C for 20 s, 55 °C for 30 s, and 68 °C for 90 s, with final extension at 68 °C for 5 min. In order to improve the sensitivity and reduce the amount of nonspecific PCR products, the second PCR reaction was performed using the FastStart High Fidelity PCR System Kit (Roche, Basel, Switzerland) with inner primers specific for each HCV subtype and genome region (Appendix A). Reaction was performed in 25 μL reaction tube with 2.5 μL 10× High Fidelity Reaction Buffer, 0.5 µL dNTP mix, 0.2 μL forward primer (50 μM), 0.2 μL reverse primer (50 μM), 0.5 μL enzyme mix FastStart High Fidelity (5 U/μL), 20.35 μL water, and 1 μL first-round PCR products used as templates. Reaction mixtures were incubated for 2 min at 95 °C followed by 45 cycles at 95 °C for 30 s, 60 °C for 30 s, and 72 °C for 60 s, with final extension at 72 °C for 7 min. Sanger sequencing of the amplified regions was performed using the BigDye^®^ Terminator v3.1 Cycle Sequencing Kit (Applied Biosystems, Waltham, MA, USA) and the previously used inner primers (Appendix A). Sequencing reaction was performed according to manufacturer’s protocol following the program: initial denaturation at 96 °C for 1 min followed by 25 cycles at 96 °C for 10 s, 50 °C for 5 s, and 60 °C for 4 min. The sequencing products were purified and analysed on the ABI Prism 3130xl Genetic Analyzer (Applied Biosystems, Waltham, MA, USA).

### 4.3. HCV Resistance Analysis

Nucleotide sequences of the sense and antisense strand were aligned and edited in *Vector NTI* v.11.5 software (Thermo Fisher Scientific, Waltham, MA, USA), while the occurrence of RAS was determined using *geno2pheno [hcv]* algorithm according to the reference sequences for every HCV subtype [38]. Briefly, all detected substitutions were categorized in three resistance classes: resistant (well-characterized, clinically relevant resistance-associated substitutions), reduced susceptibility (substitutions associated to resistance without sufficient evidence for clinical outcome), and uncharacterized substitutions (considered susceptible to DAA therapy). Resistance to DAA was defined as the presence of a single or multiple RAS from resistant class or reduced susceptibility class, while susceptibility to DAA was defined as an absence of substitutions or presence of uncharacterized substitutions. RAS for currently approved NS3 inhibitors (boceprevir, glecaprevir, grazoprevir, paritaprevir, simeprevir, telaprevir, voxilaprevir), NS5A inhibitors (daclatasvir, elbasvir, ledipasvir, ombitasvir, pibrentasvir, velpatasvir), and nucleoside NS5B inhibitor sofosbuvir were considered relevant.

### 4.4. HCV Datasets

For phylogenetic analyses, three dataset groups were built.

The first dataset group contained all the available representative reference sequences from each HCV genotype and subtype retrieved from the International Committee on Taxonomy of Viruses [61] and 300 Croatian HCV NS5B sequences from this study. This dataset was used for evaluation of previous HCV subtyping by commercial assays. The HCV reference sequences used in this study are shown in Appendix A.

In the second dataset group, the Croatian sequence data for every genotype (1a, 1b, 3a) and region (NS3, NS5A, NS5B) was supplemented with background control sequences from GenBank obtained by BLAST search on the basis of the following inclusion criteria: (1) there was at least 90% query sequence coverage, (2) country of origin and sampling date were known and clearly established, (3) sequences were unambiguously subtyped, and (4) sequences were non-recombinant and non-clonal. Ten sequences with highest similarity score were selected for every Croatian sequence; duplicate sequences were omitted from downstream analysis. Representative reference sequences were added for subtype 1a, 1b, and 3a, respectively [61]. In addition, four subtype 1b reference sequences were included to root subtype 1a phylogenetic trees, five subtype 1a reference sequences to root subtype 1b phylogenetic trees, and two subtype 3b reference sequences to root subtype 3a phylogenetic trees (Appendix A). This dataset group was used for studying subtype-specific clades, transmission networks, and phylogenetic relationships in the context of RAS.

Third dataset group included only Croatian HCV NS3, NS5A, and NS5B sequences of each subtype which were concatenated in order to increase the precision and confidence in the inferred evolutionary relationship estimates. This dataset group was used for phylodynamic analyses. 

### 4.5. Phylogenetic Analysis

Multiple sequence alignment for all datasets was performed in *ClustalX* v.2.1 [62] and edited in *AliView* v.1.27 [63]. Maximum likelihood trees for the first and second dataset groups were constructed in *Mega* v.10.2.6. [64] under 1000 bootstrap replicates with the best-fitting nucleotide substitution model for each dataset selected according to Bayesian Information Criterion (BIC). Transmission pairs were identified in the maximum likelihood trees using *ClusterPicker* v.1.2.3. [65] with a genetic distance and bootstrap support threshold of 4.5% and 70%, respectively. Phylogenetic trees were graphically edited and annotated in *iTol* v. 6.4.2. [66].

### 4.6. Phylodynamic Analysis

Phylodynamic analysis was performed using the *BEAST* v.2.6.6 and Markov chain Monte Carlo (MCMC) analysis on the third dataset group [67]. Best nucleotide substitution model was selected according to Bayesian Information Criterion in *jModelTest* v.2.4.1 [68], while best fitting models of population growth and molecular clock were selected based on path sampling analysis implemented in *BEAST*. Three demographic models of population growth (constant population size, exponential population size, Bayesian skyline) and three models of molecular clock (strict, relaxed with an uncorrelated log normal rate distribution, relaxed with an uncorrelated exponential rate distribution) were compared as coalescent priors. Briefly, marginal likelihood estimations obtained by path sampling analysis were used for Bayes factor (BF) calculations. In accordance with Kass and Raftey (1995), only log(BF) values >2 were considered significant [69]. Evaluation of temporal signal in the dataset was performed using *Tempest* v.1.5.3 [70]. Due to the lack of sufficient temporal signal in the dataset due to limited time frame of sampling, we used the previously estimated external evolutionary rate of 1.0 × 10^−3^ substitutions/site/year obtained for most of the non-structural genes [71]. The MCMC analysis was run until sufficient parameter convergence was achieved, with 0.001 sampling rate. Convergence was assessed based on the effective sample size (ESS) after a 10% burn in *Tracer* v.1.7.2 [72]; ESS values >200 for each parameter were accepted, as they indicate sufficient sampling and chain mixing. Models that failed to converge during 200 million generations were excluded from further analysis. For every subtype group in the third dataset, the estimates of the tMRCA and substitution rate were inferred. Uncertainty in the estimates were indicated by 95% highest posterior density (95% HPD) intervals.

### 4.7. Statistical Analysis

Descriptive statistics was used to summarize the main features of the data. Qualitative data was described using frequency tables, while numerical data was described using median with first (Q1) and third (Q3) quartiles. The association between the selected variables was analysed by the Mann–Whitney test for continuous data and Pearson’s chi squared test or Fisher’s exact test for categorical data using *Statistica* v.13.5. Statistical significance was defined as *p* < 0.05.

## 5. Conclusions

In summary, we found a high overall prevalence of baseline NS3 RAS (33.0%), moderate prevalence of NS5A RAS (13.7%), and low prevalence of NI NS5B RAS (8.3%) in 300 patients with chronic HCV infection in Croatia. NS3 and NS5A RAS were more common in patients infected with HCV GT1a and GT1b compared with GT3a. Resistance associated mutations in the NS5B region were detected exclusively in patients infected with HCV GT1b. Phylogenetic tree reconstructions showed two distinct clades within the subtype 1a, clade I (62.4%) and clade II (37.6%). NS3 RAS were preferentially associated with clade I. Phylogenetic analysis demonstrated that 27 (9.0%) HCV sequences had a presumed epidemiological link with another sequence and were classified into 13 transmission pairs or clusters which were predominantly comprised of GT3a viruses. Intravenous drug use was shown to be a risk factor associated with clustering. Phylodynamic analysis highlighted an exponential increase in GT1a and GT3a effective population size in the late 20th century, which is a period associated with explosive increase in the number of IDU in Croatia. This study highlights the importance of combination regimens with a high genetic barrier to resistance and different target sites as the new standard of care for HCV treatment. A better understanding of transmission dynamics in the population of IDU is critical for the increase in effectiveness of designed HCV elimination programs.

## Figures and Tables

**Figure 1 pathogens-11-00808-f001:**
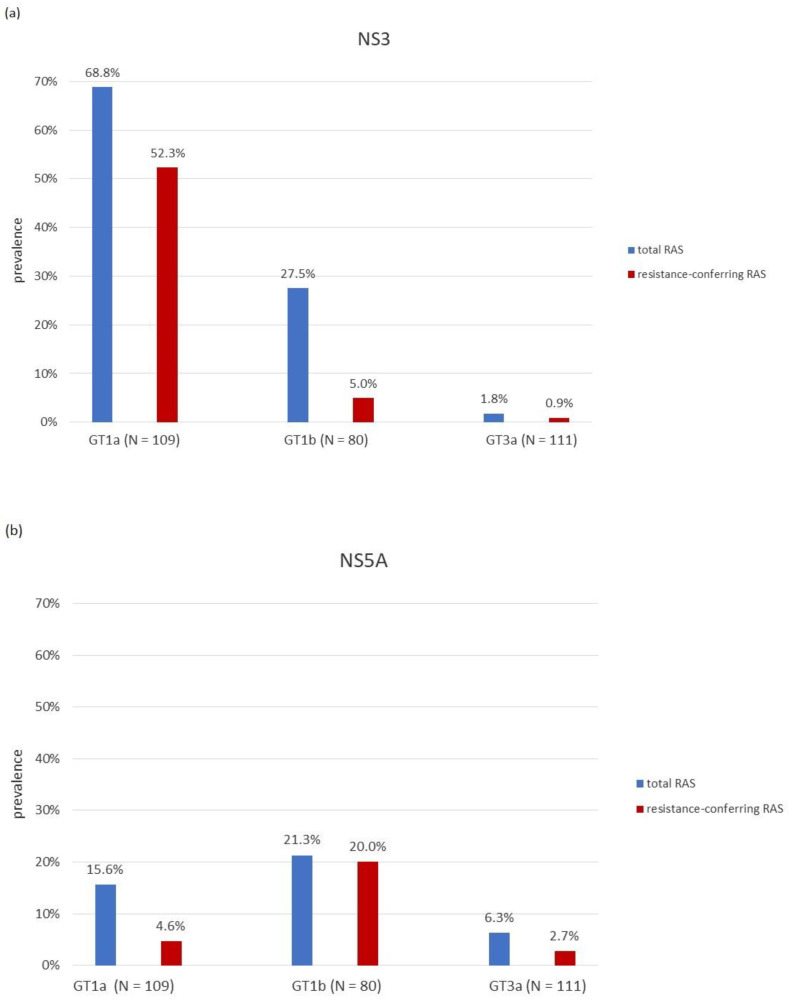
Prevalence of overall and resistance conferring RAS according to geno2pheno [HCV] algorithm and HCV genotype in: (**a**) NS3 region (**b**) NS5A region.

**Figure 2 pathogens-11-00808-f002:**
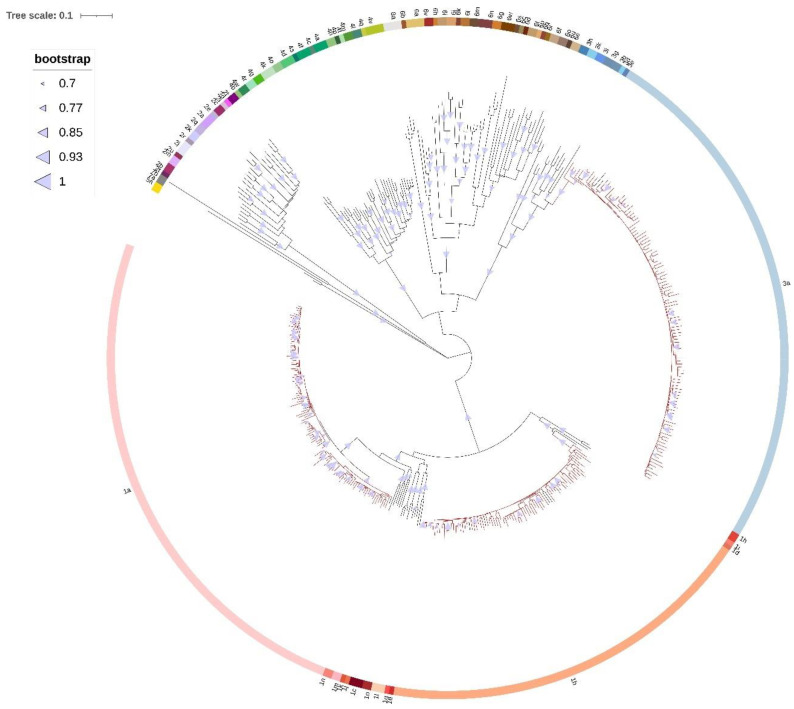
Maximum likelihood phylogenetic analysis of the HCV NS5B gene sequences constructed by applying GTR+G+I model with 1000 bootstrap replicates. Scale bar represents 0.1 nucleotide substitutions per site. Bootstrap values between 70 and 100% are displayed at the branch nodes as blue triangles with circle size corresponding to magnitude of bootstrap. Branches of reference sequences are colored black, while branches of Croatian sequences are colored red. Genotypes and subtypes are indicated by different color strips.

**Figure 3 pathogens-11-00808-f003:**
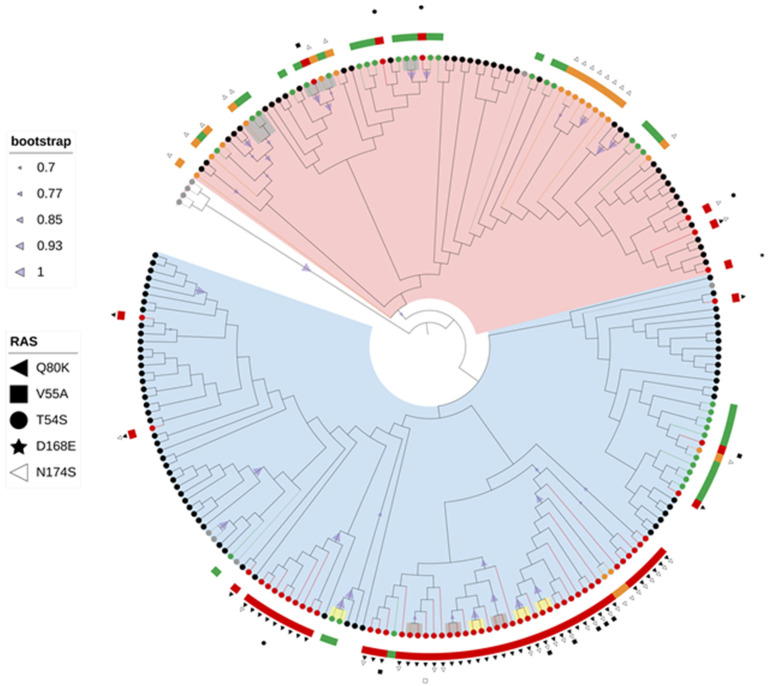
Maximum likelihood phylogenetic analysis of the NS3 gene sequences of HCV 1a subtype constructed with GTR+G+I model and 1000 bootstrap replicates. Bootstrap values between 70 and 100% are displayed at the branch nodes as blue triangles with size corresponding to magnitude of bootstrap. Branches of two most similar control sequences per each local sequence obtained by searching the BLAST database and removing duplicates are colored black. Branches of Croatian sequences without RAS are colored green, sequences with RAS conferring resistance to DAA are colored red, and sequences with RAS associated with reduced susceptibility to DAA are colored orange. Reference sequences are colored gray. Clade I sequences are highlighted blue, while clade II sequences are highlighted pink. All identified RAS are positioned on the phylogenetic tree along with the corresponding sequences. Resistance conferring RAS are marked with filled symbols, while RAS causing reduced susceptibility to at least one DAA are marked with open symbols. Identified transmission pairs are highlighted gray, while transmission pairs identified consistently across all genomic regions for GT1a are highlighted yellow.

**Figure 4 pathogens-11-00808-f004:**
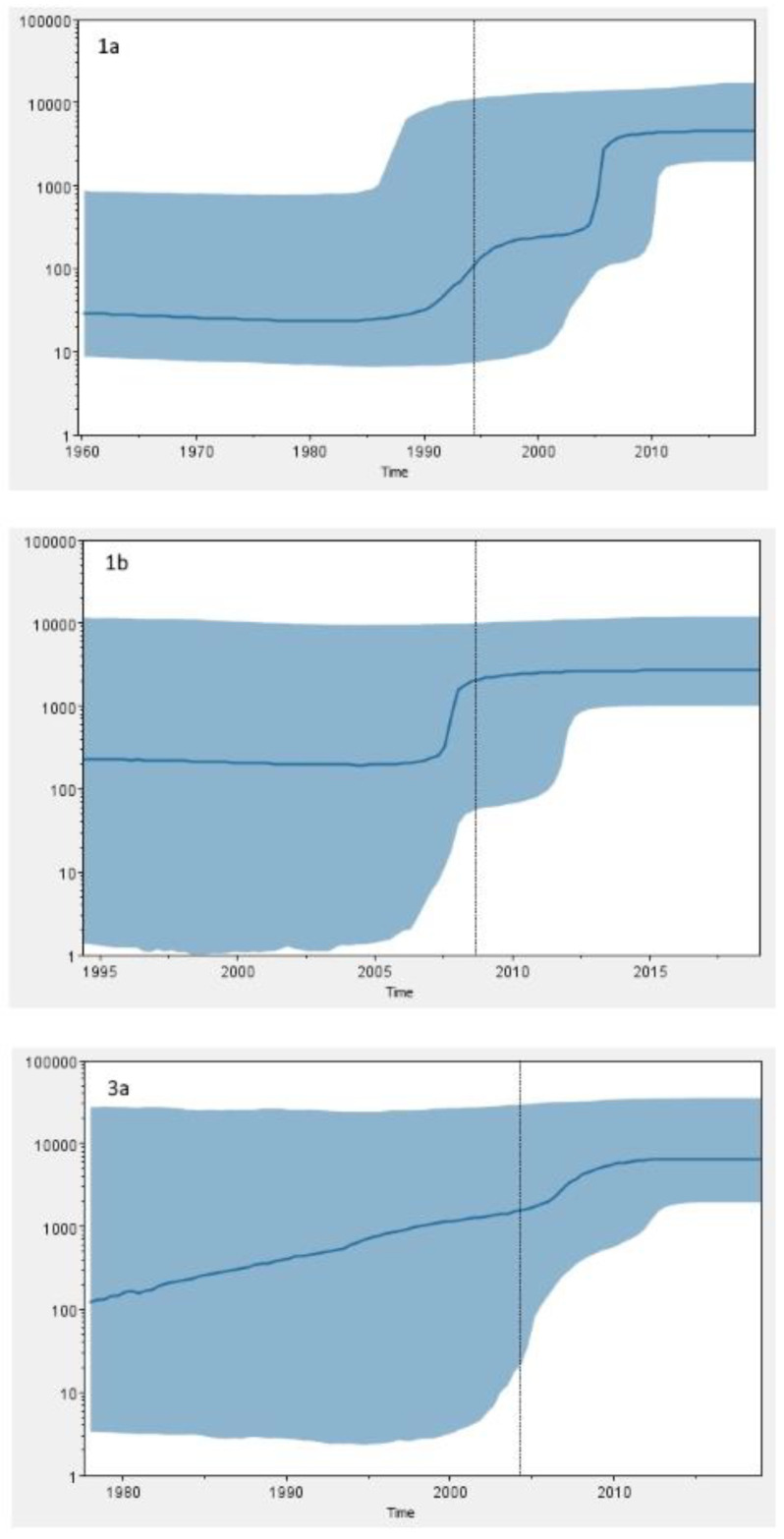
Bayesian skyline plots showing the epidemic history of the HCV subtype 1a, 1b, and 3a sequences in Zagreb, Croatia. Mean (solid blue line) and upper and lower 95% HPD (solid blue area) estimates of the effective population size (Y-axis; log_10_ scale) through time (X-axis; calendar years) from the time of the most recent common ancestor (tMRCA) are shown.

**Table 1 pathogens-11-00808-t001:** Demographic, epidemiological, clinical, and laboratory characteristics of the study population.

	Overall	Subtype 1a	Subtype 1b	Subtype 3a
Patients	N = 300	N = 109	N = 80	N = 111
Gender, *n* (%)				
M	187 (62.3)	73 (67.0)	36 (45.0)	78 (70.3)
F	113 (37.7)	36 (33.0)	44 (55.0)	33 (29.7)
Age (years), *n* (%)				
18–35	32 (10.7)	11 (10.1)	5 (6.3)	16 (14.4)
36–47	140 (46.7)	66 (60.6)	11 (13.8)	63 (56.8)
48–59	72 (24.0)	27 (24.8)	21 (26.3)	24 (21.6)
>60	56 (18.7)	5 (4.6)	43 (53.8)	8 (7.2)
Age (years), median (Q1–Q3)	45	44	61	43
	(40.0–57.0)	(40.0–50.5)	(52.0–65.8)	(39.0–49.0)
Fibrosis stage *, *n* (%)				
F0/F1	103 (35.2)	48 (44.9)	22 (27.8)	33 (30.8)
F2	72 (24.6)	27 (25.2)	21 (26.6)	24 (22.4)
F3	44 (15.0)	15 (14.0)	12 (15.2)	17 (15.9)
F4	74 (25.3)	17 (15.9)	24 (30.4)	33 (30.8)
Reported HCV infection route, *n* (%)				
IDU	95 (31.7)	48 (44.0)	1 (1.3)	46 (41.4)
Iatrogenic	48 (16.0)	14 (12.8)	25 (31.3)	9 (8.1)
perinatal/sexual	11 (3.7)	2 (1.8)	1 (1.3)	8 (7.2)
Unknown	146 (48.7)	45 (41.3)	53 (66.3)	48 (43.2)
Viral load, median (Q1–Q3), log IU/mL	6.0 (5.5–6.3)	6.0 (5.5–6.5)	5.9 (5.5–6.3)	6.0 (5.5–6.3)

Q1: first quartile; Q3: third quartile; IDU: intravenous drug use; *: available for 107 subtype 1a, 79 subtype 1b and 107 subtype 3a patients.

**Table 2 pathogens-11-00808-t002:** Comparison of demographic, clinical, and laboratory characteristics of patients with and without NS3, NS5A, or NS5B RAS.

	Presence of NS3 RAS	Absence of NS3 RAS	*p*-Value	Presence of NS5A RAS	Absence of NS5A RAS	*p*-Value	Presence of NS5B RAS	Absence of NS5B RAS	*p*-Value	Total, N (%)
Patients, *n* (%)	98 (32.7)	202 (67.3)		41 (13.7)	259 (86.3)		25 (8.3)	275 (91.7)		300 (100)
Gender, *n* (%) ^a^										
M	61 (32.6)	126 (67.4)	0.982	27 (14.4)	160 (85.6)	0.617	13 (7.0)	174 (93.1)	0.265	187 (62.3)
F	37 (32.7)	76 (67.3)		14 (12.4)	99 (87.6)		12 (10.6)	101 (89.4)		113 (37.7)
Age, median years (Q1–Q3) ^c^	44 (40–54)	46 (40–58)	0.168	46 (42.5–60.5)	45 (40–57)	0.271	57 (39–63)	45 (40–56)	0.111	45 (40–57)
HCV subtype, *n* (%) ^b^										
1a	75 (68.8)	34 (31.2)	**<0.001**	17 (15.6)	92 (84.4)	**0.007**	0 (0.0)	109 (100.0)	**<0.001**	109 (36.3)
1b	21 (26.3)	59 (73.8)		17 (21.3)	63 (78.8)		25 (31.3)	55 (68.8)		80 (26.7)
3a	2 (1.8)	109 (98.2)		7 (6.3)	104 (93.7)		0 (0.0)	111 (100.0)		111 (37.0)
Fibrosis stage, *n* * (%) ^a^										
F0/1–F2	61 (34.9)	114 (65.1)	0.278	23 (13.1)	152 (86.7)	0.609	14 (8.0)	161 (92.0)	0.920	175 (59.7)
F3–F4	34 (28.8)	84 (71.2)		18 (15.3)	100 (84.8)		11 (9.3)	107 (90.7)		118 (40.3)
Viral load, median (Q1–Q3), log IU/mL	6.0 (5.5–6.4)	6.0 (5.4–6.3)	0.303	6.0 (5.5–6.4)	6.0 (5.4–6.3)	0.849	5.9 (5.7–6.4)	6.0 (5.5–6.3)	0.682	6.0 (5.5–6.3)
Subtype 1a clade, *n* ** (%) ^b^										
1	54 (79.4)	14 (20.6)	**0.003**	12 (17.6)	56 (82.4)	0.589	0 (0.0)	68 (100.0)	1	68 (62.4)
2	21 (51.2)	20 (48.8)		5 (12.2)	36 (87.8)		0 (0.0)	41 (100.0)		41 (37.6)
Part of TC, *n* (%) ^a^										
yes	6 (22.2)	21 (77.8)	0.225	4 (14.8)	23 (85.2)	0.225	2 (7.4)	25 (92.6)	0.855	27 (9.0)
no	92 (33.7)	181 (66.3)		37 (13.6)	236 (86.4)		23 (8.4)	250 (91.6)		273 (81.0)

Q1: first quartile; Q3, third quartile; IDU: intravenous drug use; RAS: resistance-associated substitution; TC: transmission cluster or pair; ^a^: the association between the selected variables was analysed by Pearson’s chi squared test; ^b^: the association between the selected variables was analysed by the Fisher’s exact test; ^c^: the association between the selected variables was analysed by the Mann–Whitney test; *: available for total number of patients N = 293; **: available for total number of subtype 1a patients N = 109.

**Table 3 pathogens-11-00808-t003:** Detected baseline RAS to NS3 inhibitors.

Codon Position	NS3 RAS	RAS ^†^ Prevalence according to HCV Subtype		
		1a (N = 109) (%)	1b (N = 80) (%)	3a (N = 111) (%)
54	T54S	**4 (3.7)**	**2 (2.5)**	/
55	V55A	**8 (7.3)**	**1 (1.3)**	/
56	Y56F	/	18 (22.5)	/
80	Q80K	**51 (46.8)**	/	/
117	R117H	2 (1.8)	4 (5.0)	/
168	D/Q * 168E	1 (0.9)	/	/
	D/Q * 168K	/	/	1 (0.9)
	D/Q * 168R	/	/	**1 (0.9)**
174	N/S/T * 174F	/	**1 (1.3)**	/
	N/S/T * 174S	43 (39.5)	/	/
Patients with at least one NS3 RAS	any RAS	75 (68.8)	21(26.3)	2 (1.8)
	resistance-conferring RAS	57 (52.3)	4 (5.0)	1 (0.9)

RAS: resistance associated substitutions; ^†^: resistance conferring RAS are shown in **bold**; *: D168 and N174 are wild type amino acids in GT1a; D168 and S174 are wild type amino acids in GT1b; Q168 and T174 are wild type amino acids in GT3a.

**Table 4 pathogens-11-00808-t004:** Detected baseline RAS to NS5A inhibitors.

Codon Position	NS5A RAS	RAS ^†^ Prevalence according to HCV Subtype		
		1a (N = 109) (%)	1b (N = 80) (%)	3a (N = 111) (%)
24	K/Q/S * 24R	1 (0.9)	/	/
28	M/L * 28V	11 (10.1)	/	/
	M/L * 28T	**1 (0.9)**	/	/
30	Q/R/A * 30R	**3 (2.8)**	/	/
	Q/R/A * 30Q	/	**5 (6.3)**	/
	Q/R/A * 30K	/	/	**1 (0.9)**
31	L31M	**1 (0.9)**	**4 (5.0)**	/
	L31I	/	3 (3.8)	/
62	E/Q/A * 62L	/	/	4 (3.6)
93	Y93H	/	**9 (11.3)**	**2 (1.8)**
Patients with at least one NS5A RAS	any RAS	17 (15.6)	17 (21.3)	7 (6.3)
	resistance-conferring RAS	5 (4.6)	16 (20.0)	3 (2.7)

RAS: resistance associated substitutions; ^†^: resistance conferring RAS are shown in **bold**; *: K24, M28, Q30 and E62 are wild type amino acids in GT1a; Q24, L28, R30 and Q62 are wild type amino acids in GT1b; S24, M28, A30, and A62 are wild type amino acids in GT3a.

**Table 5 pathogens-11-00808-t005:** Comparison of demographic, clinical, and laboratory characteristics of patients in and out of transmission clusters.

	Patients in TC	Patients out of TC	*p*-Value	Total, N (%)
Patients	27 (9.0)	273 (91.0)		300 (100)
Gender, *n* (%) ^a^				
M	18 (9.6)	169 (90.4)	0.626	187 (62.3)
F	9 (8.0)	104 (92.0)		113 (37.7)
Age, median years (Q1–Q3) ^c^	37 (33–41)	46 (41–57.5)	**<0.001**	45 (40–57)
HCV subtype, *n* (%) ^b^				
1a	8 (7.3)	101 (92.7)	**0.008**	109 (36.3)
1b	2 (2.5)	78 (97.5)		80 (26.7)
3a	17 (15.3)	94 (84.7)		111 (37.0)
Fibrosis stage, *n* * (%) ^a^				
F0/1–F2	20 (11.4)	155 (88.6)	0.111	175 (59.7)
F3–F4	7 (5.9)	111 (94.1)		118 (40.3)
Viral load, median (Q1–Q3), log IU/mL	6.0 (4.9–6.6)	6.0 (5.5–6.3)	0.944	6.0 (5.5–6.3)
Subtype 1a clade, *n* ** (%) ^b^				
1	8 (11.8)	60 (88.2)	**0.024**	68 (62.4)
2	0 (0.0)	41 (100)		41 (37.6)
Risk factor ^a^				
IDU	14 (14.7)	81 (85.3)	**0.018**	95 (31.7)
other/unknown	13 (6.3)	192 (93.7)		205 (68.3)

Q1: first quartile; Q3: third quartile; IDU: intravenous drug use; RAS: resistance-associated substitution; TC: transmission cluster or pair; ^a^: the association between the selected variables was analysed by Pearson’s chi squared test; ^b^: the association between the selected variables was analysed by the Fisher’s exact test; ^c^: the association between the selected variables was analysed by the Mann-Whitney test; *: available for total number of patients N = 293; **: available for total number of subtype 1a patients N = 109.

**Table 6 pathogens-11-00808-t006:** Estimated time of the most recent common ancestor (tMRCA) and substitution rate (substitutions/site/year) with 95% lower and upper values from the highest posterior density (95% HPD) under various molecular clock and population growth models for HCV subtype 1a, 1b, and 3a sequences.

**GT1a**				
**Molecular Clock Model**		**Population Growth Model**		
		** *Constant* **	** *Exponential* **	** *Skyline* **
*strict*	tMRCA (year) *	1963 (1943–1981)	1961 (1939–1980)	1941 (1893–1975)
	substitution rate (10^−3^) (s/s/y) *	1.42 (0.96–1.93)	1.31 (0.84–1.76)	1.02 (0.50–1.52)
	marginal likelihood estimate (log)	−33778.19	−33691.43	−33669.42
*relaxed with lognormal rate distribution*	tMRCA (year) *	1980 (1959–1995)	1982 (1963–1988)	1960 (1911–1994)
	substitution rate (10^−3^) (s/s/y) *	2.38 (1.35–3.45)	1.99 (1.12–3.00)	1.48 (0.52–2.40)
	marginal likelihood estimate (log)	−33666.33	−33581.52	−33570.04
*relaxed with exponential rate distribution*	tMRCA (year) *	***	1996 (1984–2006)	1966 (1892–2004)
	substitution rate (10^−3^) (s/s/y) *	***	3.61 (1.84–5.69)	2.46 (0.46–4.53)
	marginal likelihood estimate (log)	***	−33629.85	−33617.95
**GT1b**				
**Molecular Clock Model**		**Population Growth Model**		
		** *Constant* **	** *Exponential* **	** *Skyline* **
*strict*	tMRCA (year) *	1994 (1985–2002)	1990 (1980–1999)	1984 (1965–1999)
	substitution rate (10^−3^) (s/s/y) *	4.00 (2.71–5.38)	3.42 (2.27–4.63)	2.86 (1.59–4.32)
	marginal likelihood estimate (log)	−32729.35	−32649.40	−32628.27
*relaxed with lognormal rate distribution*	tMRCA (year) *	2004 (1997–2009)	1998 (1986–2007)	1994 (1973–2009)
	substitution rate (10^−3^) (s/s/y) *	6.95 (4.18–9.71)	4.52 (2.30–6.80)	3.78 (1.30–6.59)
	marginal likelihood estimate (log)	−32680.06	−32595.23	−32572.04
*relaxed with exponential rate distribution*	tMRCA (year) *	***	2002 (1991–2010)	2003 (1983–2013)
	substitution rate (10^−3^) (s/s/y) *	***	7.06 (2.87–11.80)	8.24 (1.29–15.60)
	marginal likelihood estimate (log)	***	−32639.60	−32631.35
**GT3a**				
**Molecular Clock Model**		**Population Growth Model**		
		** *Constant* **	** *Exponential* **	** *Skyline* **
*strict*	tMRCA (year) *	1964 (1940–1985)	1961 (1934–1981)	1929 (1853–1978)
	substitution rate (10^−3^) (s/s/y) *	1.92 (1.15–2.70)	1.68 (1.02–2.38)	1.22 (0.48–2.02)
	marginal likelihood estimate (log)	−38484.10	−38378.81	−38362.98
*relaxed with lognormal rate distribution*	tMRCA (year) *	1993 (1978–2004)	1990 (1975–2002)	1978 (1933–2004)
	substitution rate (10^−3^) (s/s/y) *	3.85 (2.21–5.53)	2.86 (1.60–4.24)	2.04 (0.47–3.67)
	marginal likelihood estimate (log)	−38404.69	−38298.16	−38269.86
*relaxed with exponential rate distribution*	tMRCA (year) *	***	1999 (1987–2008)	***
	substitution rate (10^−3^) (s/s/y) *	***	4.87 (2.18–7.77)	***
	marginal likelihood estimate (log)	***	−38342.03	***

*: numbers in brackets indicate 95% lower and upper values from the highest posterior density (95% HPD); ***: model did not converge.

## Data Availability

All 300 NS3, 300 NS5A, and 300 NS5B sequences obtained in this study were submitted to GenBank nucleotide sequence database under accession numbers OM312065–OM312964.

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
