# Peer review of "Molecular Epidemiology and Baseline Resistance of Hepatitis C Virus to Direct Acting Antivirals in Croatia"

_pathogens, 2022, doi:10.3390/pathogens11070808_

Round 1

Reviewer 1 Report

Petra Simicic et al. assessed the characteristics of HCV RAS at NS3, NS5A and NS5B in 300 Croatian subjects. The article was well written and can provide an important insight into the molecular epidemiology of HCV in Croatia. Only minor comments are raised in the text.

1. 2.1 Study population: please trim the unnecessary wordings (line 93) (109/300) and just state 36.3%. Furthermore, add the percentages of GT1b and 3a in lines 94-95. Same in lines 99-101.

2. 2.1.1 Prevalence of NS3 specific RAS: line 125 (99/300) is not consistent with Table 2 to be 98/300; line 128 (1b, 22/80) is not consistent with Table 2 to be 21. Please clarify it.

3. How did the authors manage patients with mixed GT infections? Did the Sanger sequencing work in those with mixed infections? Would this study potentially underestimate the possibility of mixed infection because a substantial proportion of patients had IDU or unsafe medical procedures?

Author Response

Reviewer 1

  1. 1 Study population: please trim the unnecessary wordings (line 93) (109/300) and just state 36.3%. Furthermore, add the percentages of GT1b and 3a in lines 94-95. Same in lines 99-101.

The unnecessary wordings were trimmed in line 93 and lines 99-101. Percentages of GT1b and GT3a in lines 94-95 were added.

  1. 1.1 Prevalence of NS3 specific RAS: line 125 (99/300) is not consistent with Table 2 to be 98/300; line 128 (1b, 22/80) is not consistent with Table 2 to be 21. Please clarify it.

Prevalence of NS3 specific RAS is correctly stated in Table 2. Presence of NS3 RAS was detected in 98 patients overall, and in 21 patients infected with HCV GT1b. Therefore, the manuscript text in lines 125 and 128 was corrected to 32.7% (98/300) and 26.3% (21/80), respectively. Furthermore, the same correction was made in Table 3 for GT1b.

  1. How did the authors manage patients with mixed GT infections? Did the Sanger sequencing work in those with mixed infections? Would this study potentially underestimate the possibility of mixed infection because a substantial proportion of patients had IDU or unsafe medical procedures?

HCV genotype was previously determined by VERSANT HCV Genotype 2.0 Assay LiPA which is a second-generation line probe assay containing subtype-specific primers probes targeting both the 5’ UTR and the core regions of the viral genome. According to this genotyping method, none of the patients had mixed GT infections. However, this assay is not designed for confident identification of mixed infections, therefore we conducted supplementary HCV genotyping based on phylogenetic analyses by sequencing NS5B region of the viral genome since it is well recognized that NS5B is the most discriminant region for HCV genotyping. Maximum likelihood phylogenetic tree confirmed that all 300 Croatian sequences were correctly subtyped with no mixed GT infections. Similarly, Chen et al. (2017) found mixed infections of the dominant HCV subtypes in a very small portion of samples (n=65, 0.203%) in a large cohort of patients based on real-time PCR and Sanger sequencing of core/envelope 1 region. Goletti et al. (2019) showed that the global agreement between VERSANT LiPA assay and Sanger sequencing was greater than 95%. However, J.A. del Campo et al. (2018) demonstrated that the routine method (LiPA) failed to accurately subtype seven out of 84 infections (8.5%), while deep sequencing techniques enabled detection of six mixed infections in 84 samples (7%). It seems that subtyping analysis based on deep sequencing enables the most reliable analysis of HCV genome diversity with the added benefit of the possibility of detecting mixed infections, but with significantly higher cost compared to routine methods. Therefore, we briefly mentioned this in lines 486-487 as one of the limitations of our study.

Previous studies have shown that multiple HCV infections such as coinfenctions, mixed infections and superinfections are common in IDU due to the specific modes of transmission characteristic for this population. Shooting galleries, where IDU can rent or borrow needles and syringes, are a high-risk environment for HCV transmission. However, since the beginning of the 21st century Croatia has implemented harm reduction programs which aim to minimize the negative impacts associated with drug use. Free replacement of syringes and needles is one of the most important aspects of harm reduction. Low prevalence of mixed HCV infections in Croatia, even in IDU population, could partly be attributed to successful implementation of these programs.

References:

Chen Y, Yu C, Yin X, Guo X, Wu S, Hou J. Hepatitis C virus genotypes and subtypes circulating in Mainland China. Emerg Microbes Infect. 2017 Nov 1;6(11):e95. doi: 10.1038/emi.2017.77. 

Goletti S, Zuyten S, Goeminne L, Verhofstede C, Rodriguez-Villalobos H, Bodeus M, Stärkel P, Horsmans Y, Kabamba-Mukadi B. Comparison of Sanger sequencing for hepatitis C virus genotyping with a commercial line probe assay in a tertiary hospital. BMC Infect Dis. 2019 Aug 22;19(1):738. doi: 10.1186/s12879-019-4386-4. 

Del Campo JA, Parra-Sánchez M, Figueruela B, García-Rey S, Quer J, Gregori J, Bernal S, Grande L, Palomares JC, Romero-Gómez M. Hepatitis C virus deep sequencing for sub-genotype identification in mixed infections: A real-life experience. Int J Infect Dis. 2018 Feb;67:114-117. doi: 10.1016/j.ijid.2017.12.016.

Pham ST, Bull RA, Bennett JM, Rawlinson WD, Dore GJ, Lloyd AR, White PA. Frequent multiple hepatitis C virus infections among injection drug users in a prison setting. Hepatology. 2010 Nov;52(5):1564-72. doi: 10.1002/hep.23885. 

Cunningham EB, Applegate TL, Lloyd AR, Dore GJ, Grebely J. Mixed HCV infection and reinfection in people who inject drugs--impact on therapy. Nat Rev Gastroenterol Hepatol. 2015 Apr;12(4):218-30. doi: 10.1038/nrgastro.2015.36. 

Other remarks

We have observed that the first two subheadings of the “2. Results” section have the same subheading numbers (“2.1. Study population” and “2.1. Resistance analysis”). Therefore, the subheading number of the second subheading was corrected to “2.2. Resistance analysis”. All further subheading and subsection numbers in the “2. Results” section were corrected accordingly.

Reviewer 2 Report

T
he authors of the manuscript investigated molecular epidemiology and baseline resistance of HCV to DAAs in Croatia. The topic is interesting, although since the introduction of highly effective DAAs it has had limited practical significance.

There are errors in the manuscript that should be corrected

1)     The structure of the manuscript is not correct: The '"Material and method" section is located after "Discussion"

2)     The name of the cited author in the Discussion was incorrectly written (Parziewski instead of Parczewski)

3)    The article would require minor language changes

Author Response

Reviewer 2

  1. The structure of the manuscript is not correct: The '"Material and method" section is located after "Discussion"

The manuscript was prepared according to the instructions for authors for MDPI Pathogens journal and the provided Microsoft Word template (available at:  https://www.mdpi.com/journal/pathogens/instructions). Both manuscript preparation guidelines and the provided template have the “Materials and Methods” section located after the “Discussion” section and before the “Conclusions” section.

  1. The name of the cited author in the Discussion was incorrectly written (Parziewski instead of Parczewski)

The name of the cited author was corrected to Parczewski instead of Parziewski.

  1. The article would require minor language changes

We have proofread the manuscript again and corrected typographical errors and mistakes in grammar, style, and spelling. All changes are indicated using the “Track Changes” function of MS Word.

Other remarks

We have observed that the first two subheadings of the “2. Results” section have the same subheading numbers (“2.1. Study population” and “2.1. Resistance analysis”). Therefore, the subheading number of the second subheading was corrected to “2.2. Resistance analysis”. All further subheading and subsection numbers in the “2. Results” section were corrected accordingly.
